# The Sacred Writing of Knowledge: Interpreting the True Form Charts of the Man-Bird Mountain in Taoism

## Linbo Cai

Department of Philosophy, East China Normal University, Shanghai 200062, China; linbo1969@sina.com

**Abstract:** The True Form Charts of the Man-Bird Mountain (人鳥山真形圖, or TFCMBM) are important ritual images created by ancient Taoists. Scholars envisage them as the "fairy mountain map" (仙山圖) or "fairyland picture" (仙境圖) imagined by ancient Taoists. However, TFCMBM is actually a description of the mechanism of the "convergence of celestial and terrestrial *qi*"(天地氣交) in the human body. According to ancient Chinese medicine, the mechanism of the inter-induction of *qi* (氣交) controls the transmission and transformation of food, circulation of *qi*-blood in the body, and generation and storage of vital essence. Ancient Taoist priests referred to the mechanism of the inter-induction of *qi* as "regulating qi in Central Yellow" (黃中理炁) and as a kind of image, "the true shape of the human bird mountain" to guide novice Taoists in meditation (存思) and activating *qi* (行氣). This study argues that the TFCMBM is a portrayal of the mechanism of the inter-induction of *qi* and reveals the Taoist method of writing sacred knowledge and its cognitive nature.

**Keywords:** Taoism; the True Form Charts of the Man-Bird Mountain; convergence of celestial and terrestrial *qi*; sacred writing of knowledge

## 1. Introduction

Historically, Taoist priests have often adopted mystical images and written symbols to express their thoughts and knowledge. Deciphering and interpreting the implicit meaning of Taoist symbols has always been a challenge for researchers. This study's interpretation of the True Form of the Human Bird Mountain (人鳥山真形圖), or TFCMBM, may provide an enlightened perspective and method to solve this problem.

As important Taoist ritual images, the TFCMBM have long been of interest to scholars. Researchers generally regard them as a combined map of an "immortal mountain" (仙山) or "fairyland" with theological and geographic significance. Some scholars directly regard the TFCMBM as the Kunlun Mountains, believing that the "Kunlun Mountain is the Man-Bird Mountain or Spirit-Bird mountain, and the mother of birds is the mother of the West" (Wang 2001, p. 321) or assert that "Man-Bird Mountain" (人鳥山) is the sacred derivative of the Kunlun Mountains (Kunlun Mountain) (Li 2002, pp. 99–103). Other scholars have explored the essence of the material composition of the "Man-Bird Mountain", emphasizing that the "Man-Bird Mountain" is the True Form Charts of the universe composed of original *qi* (元氣) (Lagerwey 1997, p. 79). They have also emphasized the description of its cloud, gas, and water flow patterns, calling it the "True Form" (真形) (Zhan 1997, pp. 41–42), which reflects a fairyland rich in natural resources (Huang 2014, p. 129). These viewpoints focus on the external wonderland of the "Man-Bird Mountain" but do not reveal the internal structure of the TFCMBM.

Significantly, some scholars have discovered that the TFCMBM may hint at the human body's internal structure. One important clue is that the two paragraphs attached to the edge of the *Picture of the Scripture of the Mystic Vision of the Man-Bird Mountain (Xuanlan renniao shan jing tu* [玄覽人鳥山經圖]; see Figure 2) are derived from the description of the two mountains: *Taiyuan* (太元) and *Changgu* (長穀) in *Baopuzineipian weizhi* (抱樸子內篇·微旨) written by Ge Hong (葛洪). Accordingly, Li Fengmao suggested that Ge Hong's 葛洪 "text

is also an extremely valuable document in the ancient Taoist books on Taoist meditation, and the two mountains are not from outside but only in the essence of thought…". Sweet wine (醴泉) and Polygonatum (黃精), described in the text, can be used as metaphorical imagery to symbolize the immortal goods of the Taoist wonderland; they may also represent a personal experience that is felt throughout one's body (Li 2019, pp. 317–18). Likewise, he believed that "the True Form Charts of famous ancient mountains and the Man-Bird Mountain is also meditative in nature (Li 2009, p. 174). Gil Raz stated more clearly that "We may add here that the amalgamation of these two mountains in the man-bird chart once again reveals the cosmic nature of Mount Man-bird which, such as the Dao, unifies yin and yang within itself" (Raz 2010, pp. 1440–41). Unfortunately, the two scholars did not examine this issue in depth. It should be noted that Fabrizio Pregadio, in a recent article, profoundly pointed out that the True Form of the Man-Bird Mountain is not only the "true form" of the macrocosm but also the "true form " of the human microcosm (Pregadio 2020, pp. 66–67).

Accordingly, this study builds on the results of previous research and provides an in-depth investigation of the TFCMBM, particularly to make a breakthrough interpretation of the metaphor underlying the two mountains, Taiyuan (太元) and Changgu (長穀), to reveal the inner structure of the TFCMBM and its related intellectual connotations.

## 2. Creation of the TFCMBM

Academics generally believe that the TFCMBM was probably created between the time of the Northern and Southern dynasties up until the Tang Dynasty, but there are differences in the specific period. Some scholars argue that the images originated during the Tang Dynasty, while others maintain that they derived from the Northern and Southern dynasties.

In The Taoist Canon: A Historical Companion to the Daozang (道藏), Kristofer M. Schipper and Franciscus Verellen list the two extant copies of the TFCMBM—Picture of the Scripture of the Mystic Vision of the Man-bird Mountain (玄覽人鳥山經圖) (referred to as Picture of the Mystic Vison, 玄覽圖) and the Primordial View of the Mountain Form of the Man-Bird Mountain (Yuanlan shan jing tu, 元覽山經圖, referred to as the Picture of the Primordial View, 元覽圖)—as the classics of the Tang Dynasty (Vitiello 2004, p. 421). Other scholars have found that the Picture of the Mystic Vision contains influential elements from Indian Tantric mantra images; therefore, the TFCMBM must have appeared during the Tang Dynasty. For example, according to Xin Deyong (辛德勇), the Picture of the Mystic Vision is "surrounded by explanatory words, obviously with traces of imitating the Buddhist Tantric (Tuoluoni) mantras" (Xin 2016, p. 7). Shi-shan Huan (黃士珊) affirmed thus: "The text-image juxtaposition resembles the single-sheet design of Buddhist charms known as the Dhāranī Chart of the True Word (Tuoluoni zhengyan, 陀羅尼真言), which were popular in the 9th and 10th centuries" (Huang 2012, p. 139). Additionally, "The fact that Daoists borrowed the Buddhist design for their true form charts suggests that the extended charts had a similar function and were probably created in the same period" (Huang 2012, p. 139). This view appears to have some merit. The text of the Picture of the Mystic Vision includes foreign objects such as the Buddhist "Sumeru" (須彌山) and the "zhentan spice" (震檀香) of the Yuezhi State (月支國). The schema is called "the mantra of the human bird mountain" (人鳥山經咒; Dongzhen taishang badao mingji jing 洞真太上八道命籍經 1988, p. 512); it is similar to the incantation texts of Esoteric Buddhism (密教, guhya). However, given the extraterritorial "sacred mountains" (including South and Central Asia) and related names in China's theological geography system, the mantra was an intellectual trend in Eastern Jin and Northern dynasties. In fact, during the Six Dynasties, Taoists already regarded the Buddhist "Sumeru mountain" (須彌山) as an immortal Taoist site and even regarded "Sumeru mountain" and the Man-Bird Mountain as the same location. For example, there was a popular legend in Taoism at the time. In Kapilavastu, the hometown of Buddha Sakyamuni, there was a "Mount of Sumeru of Spirit Flying Man-Bird" (須彌靈飛人鳥之山), embodied by the overseas administration of the Primeval Taoist King 元始天王 (Wushang Miyao, 無上秘要, Yu 1988, p. 63). On this mountain, there were Seven Treasures (sapta ratnani 七寶) palaces and high steps in heaven. The Queen Mother of the West 西王母

also climbed this mountain when she first learned about Taoism. The mountain is also known as the "Kunlun Mountain of Man-Bird" (昆侖人鳥之山). There are complete Taoist sandong (三洞) classics in the mountain, each word of which is approximately 10 feet long. The most important text among them is the True Form Charts of Kunlun Mountain of Man-Bird (昆侖人鳥山之真形圖); if a person is able to obtain these charts, they can become immortal in the Three Caverns (三洞玄真 Shangqing taiji yinzhu yujing baojue 上清太極隱注玉經寶訣 1988, p. 644). During the Six Dynasties, Taoists integrated various cultural elements of Buddhism, such as Kapilavastu, Sumeru, and sapta ratnani, into the concept of the Man-Bird Mountain of Taoism. Hence, it is not true that the earliest appearance of the TFCMBM occurred during the Tang Dynasty.

Increasingly many scholars have come to believe that the TFCMBM originated in Eastern Jin and Northern and Southern dynasties. Jiang Sheng (姜生) suggested that the TFCMBM were created in the Northern and Southern dynasties (Jiang and Tang 2010, pp. 882–83). Luo Yiying (Luo 2010, p. 137) carefully collated the relevant literature and conducted in-depth research. She found that at the end of the Eastern Jin Dynasty, the classical Taoist *Dongxuan wufu jing* (洞玄五符經), *Shizhouji* (十洲記), *Basu zhenjing* (八素真經), *Ciyi jing* (雌一經), and *Taishang yujing yinzhu* (太上玉經隱注) recorded the content of the TFCMBM and can be reflected in the *Picture of the Mystic Vision* (玄覽人鳥山經圖). Accordingly, she asserted that the book, *Picture of the Mystic Vision* (玄覽人鳥山經圖), was written between the middle and late Eastern Jin Dynasty to the early Tang Dynasty and may be dated before 668 AD at the latest (Luo 2010, p. 137). Regarding this conclusion, although the timespan is too broad, the TFCMBM may have appeared as early as the middle and late Eastern Jin Dynasty. Giovanni Vitiello also observed that parts of the scriptures in the *Picture of the Mystic Vision* (玄覽圖) are quoted from the *Dongxuan wufu jing* (洞玄五符經) of the Eastern Jin Dynasty and thought that the *dongxuanwufujing* (洞玄五符經) might be the lost *Wufu renniao jing* (五符人鸟经) (Vitiello 2004, p. 422). Of interest is the research of Wang Haoyue (王皓月) and Bai Zhaojie (白照傑, unpublished) in this area. Wang Haoyue (王皓月) stressed that the *Five Talismans of Man-Bird* (人鳥五符), compiled by Lu Xiujing (陸修靜) of the Southern Dynasty, probably includes the TFCMBM; this work has been widespread since the Liu Song Dynasty (pp. 420–79) (Wang 2017, pp. 143–69). Bai Zhaojie (白照傑) affirmed this view and noticed visual symbols (see Figure 1) in the fragments of the *Dunhuang Book of Taogongchuanyi* (陶公傳儀), which is painted with symbols such as a *bird* (鳥), *seal* (印), *water* (水), *prohibition* (勿), and *nine* (九). Bai Zhaojie (白照傑) speculated that this picture was probably a symbolic version of the *Talismans of Man-Bird Mountain* (人鳥山符); that is, the TFCMBM (Bai, unpublished). This symbol was very popular for use during Taoist rites during the Tang Dynasty. Taoist Zhang Wanfu (張萬福) recorded such a process of inviting immortals in a text of the ceremony:

> Respectfully invite: True gods and their mounted aides in the form of the Man-Bird Mountain, willing to come from heaven to earth and sit on the seat of *jiao* (醮). 謹奉請: 人鳥山形真靈騎從, 一合來下, 降臨醮席. (Zhang 1988, pp. 429, 495)

The "form of the Man-Bird Mountain" is probably a nickname for the TFCMBM. It refers to the "true gods" that Taoists invited to participate in the rites, who were originally "living" in that "picture", that is, in the TFCMBM.

All indications are that the TFCMBM is likely derived from China's own ancient cultural traditions. Huang traced the history of the painting of the Man-Bird (人鳥) and argued that the Man-Bird (人鳥) on the T-shaped silk painting from the Han tomb of Mawangdui (馬王堆) and the Man-Bird (人鳥) of Laojun (老君) from the 2nd century AD reflects the tradition of painting the Man-Bird (人鳥圖) in early Chinese art. Taoism drew upon this ancient tradition of drawing the Man-Bird (人鳥) and transformed it into something immortal (Huang 2012, pp. 142–43).

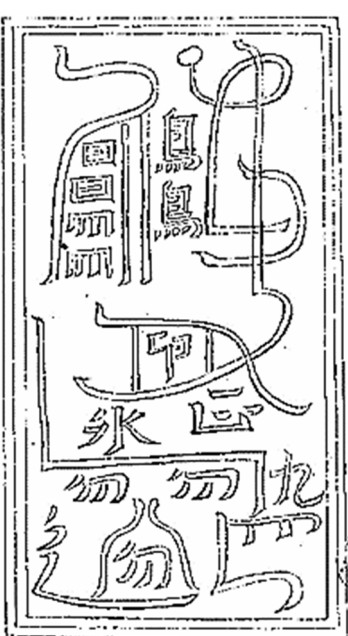

**Figure 1.** *Talismans of Man-Bird Mountain* (人鳥山符, Li 1999, p. 2638). [Photo reprinted with permission from *Dunhuang Daozang, Vol. 5*, (敦煌道藏, 第5 冊) edited by Li Defan(李德範), Beijing: China National microfilming center for library resources. 1999. p. 2638. This picture has no copyright issues].

This investigation reminds us that although the TFCMBM are mixed with foreign cultural factors such as mainstream and Esoteric Buddhism, their inner structure and ideological thinking should form the continuation of China's cultural traditions. The appearance of the TFCMBM is due to the historical evolution of ancient Chinese culture rather than the stimulation of foreign religions. Hence, whether from textual evidence or historical logic, the conclusion that the TFCMBM was created in the middle and late Eastern Jin dynasties appears reasonable.

### 3. Textual Interpretations of the TFCMBM

In the extant Taoist literature, there are two versions of the TFCMBM: (1) Chart of the True Form of the Topography of the Most High Man-Bird Mountain (太上人鳥山真形圖), which is included in the Picture of the Scripture of the Mystic Vision of the Man-Bird Mountain (玄覽人鳥山經圖; Picture of the Mystic Vision: 玄覽圖) in the dongxuan department (洞玄部) of zhengtongdaozang (正統道藏) during the Ming dynasty; (2) the TFCMBM, which are included in the Primordial View of the Mountain Form of the Man-Bird Mountain (元覽人鳥山形圖; Picture of the Primordial View: 元覽圖) in the Taixuan department's (太玄部) 80th volume of yunjiqiqian (雲笈七箋) of zhengtongdaozang (正統道藏) from the Ming dynasty.

#### 3.1. Picture of the Mystic Vision (玄覽圖)

*Picture of the Mystic Vision* (玄覽圖) consists of words and images. The words primarily describe the theological imagery of the TFCMBM, how the ritual is conducted, and its functions. The combined image is displayed in a square format of clouds and air, similar to the True Form Charts of the Five Sacred Peaks (五嶽真形圖). The surrounding border has two circles of text (寄文; see Figure 2).

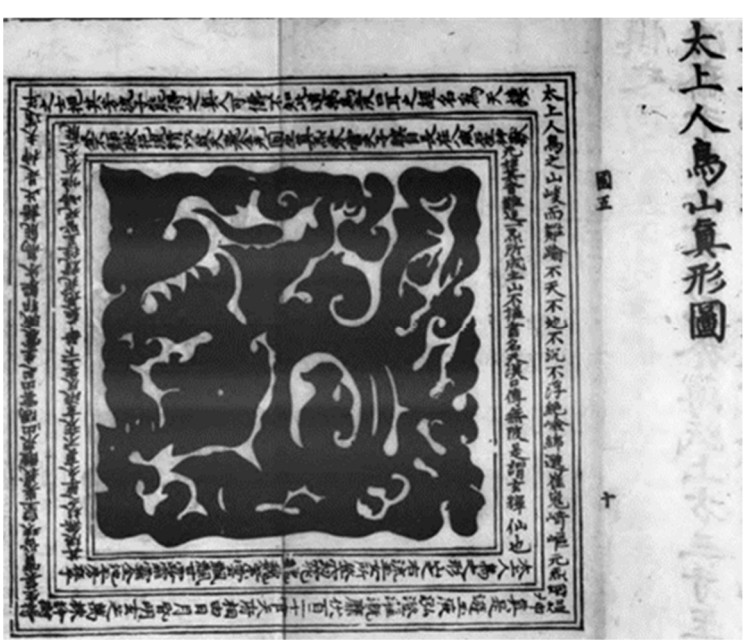

**Figure 2.** *Chart of True Form of the Topography of the Most High Man-Bird Mountain* (太上人鳥山真形圖, Xuanlan renniaoshan jing tu 玄覽人鳥山經圖 1445). [Photo reprinted from *Zhengtongdaozang* 正統道藏, Official Publication of the Ming Dynasty, 10th year of the Zhengtong era. This picture has no copyright issues].

Figure 2′s attached-text below (122 words) describes the outer circle (Huang 2012, pp. 147–48).

| | |
|---|---|
| The Most High Man-Bird Mountain, | 太上人鳥之山 |
| Is so high it is hard to climb. | 峻而難踰 |
| It is neither heaven nor earth, | 不天不地 |
| Neither sinking nor floating; | 不沉不浮 |
| Extremely dangerous and continuously following, | 絕險綿邈 |
| It has magnificent and rugged mountains. | 崔嵬崎嶇 |
| In the rising mist of primordial *qi*, | 元炁煙熅 |
| The perfected is roaming. | 神真是遊 |
| Jade liquid [saliva], abundant and pure, | 玉液泓澄 |
| Irrigates the body continuously. | 灌溉靡伏 |
| One hundred and twenty celestial officers | 百二十官 |
| Follow one another in the heavenly office. | 天府相由 |
| The sun and moon are dim and bright, | 日月昏明 |
| Mysterious ganoderma lucidum and gracilaria, | 玄芝蘺株 |
| For scarlet trees to grow in particular, | 絳樹特生 |
| And their fruit are like pearls. | 其實皆珠 |
| The white-jade [cliffs] are craggy, | 白玉嵯峨 |
| Sweet-wine springs flow from crevices, | 醴泉出隅 |
| Ganoderma lucidum grow, clouds appear, | 雲出芝生 |
| Thunder dies away, evil scatters, | 震滅邪軀 |
| Trees and birds know to speak, | 木鳥能語 |
| All because of the sacred power. | 此是神夫 |
| Gentlemen who have returned to youth, | 還年之士 |
| Fetch from its [the mountains] fragrant streams; | 挹其芳流 |
| If you can get here, | 子能得之 |
| You become a companion of the perfected. | 真人可儔 |
| If you do not know this Dao, | 不知此道 |
| Like the myriad things you return to dust and ashes. | 萬為土灰 |
| In the transmission from mouth to ear [oral], | 口耳之經 |
| It is called the Celestial Tower. | 名為天樓 |

The text below (122 words) describes the inner circle (Huang 2012, pp. 148–49).

| The form of the Most High Man-Bird, | 太上人鳥之形 |
| Flowing from the right side of the mountain. | 山之右流 |
| Where the jade maidens climb up. | 玉女所登 |
| Fair figures ascending the rising summit, | 窈窕巍巍 |
| Purple clouds are floating, | 紫雲飄飄 |
| Sweet dew falls abundantly, | 甘露霏霏 |
| Golden ponds and jade rooms, | 金池玉房 |
| Are at the corners of the mountains. | 在乎其隈 |
| Branchless grass, | 無枝之草 |
| Never withers, in summer or winter; | 冬夏不衰 |
| Its mysterious sap, once exhausted, will regenerate, | 玄液反生 |
| Covering the grass with flowery lushness. | 上下華蕤 |
| The ignorant masses compete to go, | 愚兆兢往 |
| But as soon as they are there, they will all die. | 至皆死歸 |
| Only the Highest Lord | 惟有太上 |
| Can climb without crumbling, | 能登不頹 |
| Pour out flowing essence and rinse his mouth with it, | 漱挹流精 |
| Hence he is able to fly in heaven; | 以致天飛 |
| Golden rays form a halo, | 金光圓生 |
| The perfected *qi* rides thunder. | 真炁乘雷 |
| The master closes his eyes, | 夫子瞑目 |
| Focusing on the eight mighty dragons continuously. | 長在八威 |
| Divine animals of primordial origin born from stones, | 石生神獸元始 |
| So their sound is difficult to trace. | 其音難追 |
| Formed by pure *qi*, | 一炁所成 |
| The earth mountain never collapses. | 土山不摧 |
| This book is called the Heavenly River, | 書名天漢 |
| And it will be transmitted orally far and wide. | 口傳無陂 |
| This is the mysterious light, | 是謂玄輝 |
| The immortal. | 仙也 |

The text about the outer circle is clear and unambiguous. However, there are questions about the sentences and meaning of the text about the inner circle.

(1) The word *primeval* (元始) is incoherent with both the preceding and following sentences. Huang placed *primeval* (元始) after "Divine animals of primordial origin born from stones" (石生神獸), and the text does not make sense. The phrase *divine animals* refers to *eight mighty dragons* (八威), not *primeval* (元始). The phrase *eight mighty dragons* (八威神獸) is found in the Taoist scripture Taishang daoyin sanguang jiubian miaojing 太上導引三光九變妙經 (1988, p. 857). Likewise, *primitive* (元始) cannot be connected with *their sound, which is difficult to trace* (其音難追). John Lagerwey affirmed that after primeval (元始), "and no end?—two characters are missing in this inscription" (Lagerwey 1987, p. 165). However, from the figure, there is no vacancy before or after *primitive*. Therefore, Lagerwey's view is not tenable.

(2) *The immortal* (仙也) is the last phrase of the text about the inner circle, which cannot be connected with the previous phrase, *This is the mysterious light* (是謂玄輝). If these phrases were forcibly connected, they would not conform to ancient Chinese habits of expression. We can see that there is no vacancy before or after *The immortal* (仙也); therefore, there is no problem with missing words.

Then why does the text about the inner circle have these issues? The real reason is that it conceals a special method of reading sentences involving palindromes (回文). Regarding the "sentences" and "words that count" of the text about the inner circle, the scripture of the *Picture of the Mystic Vision* suggests thus:

There are 11 natural words inside the mountain. Abbreviated secret instructions [attached text] for oral recitation are included here: [when] Arcuate Dragons move around [and] deities manifest themselves, dreadful calamities will be eliminated and the ominous will be dispelled. The characters in the void inscribed outside the mountain include 124 facing left and 120 facing right. Together, there are

244 characters ([Huang 2012](), pp. 151–52). 山內自然之字一十有一，其訣口中寄文附出：弓龍行、神出，除凶殃、辟非祥。山外空虛之字：向左百二十四，向右百二十，合二百四十四字。([Xuanlan renniaoshan jing tu 玄覽人鳥山經圖 1988](), p. 698)。

In this case, the "characters in the void inscribed outside the mountain" (山外空虛之字) appear to refer to two circles of text on the border of *the Picture of the Mystic Vision* (玄覽圖). The total number of 244 words is the same as that described in the scripture because the diagram indicates that both the inner and outer circles of the text amount to 122 words.

However, the scripture also says, *To the left 124, and to the right 120*, which is inconsistent with the number of words in the two circles of text (122 words). Why is that? This sentence should refer to the reading of the text about the inner circle. The first sentence about the inner circle says: *The form of the taishang Man-Bird is the right stream of the mountain* (太上人鳥之形, 山之右流). The *right stream* (右流) refers to *flowing from right to left* (see Figure [3]): (1) starting from the lower right (northwest, representing the underground); (2) going toward the upper right (southwest, denoting heaven); (3) to the left (east); and (4) back to the starting point (northwest). It is an anticlockwise rotational movement called levorotation (左旋). In the *Picture of the Mystic Vision* (玄覽圖), it represents the terrestrial *qi* (地氣) rising into heaven. It means that the direction of the sentences in the text about the inner circle should also be consistent with it; that is, the text should be read anticlockwise (124 words):

| | |
|---|---|
| The Most High immortal, | 太上仙也 |
| Is called mysterious light, | 是謂玄輝 |
| His pithy formula will be transmitted far and wide. | 口傳無陂 |
| This book is called the Heavenly River, | 書名天漢 |
| It is buried in the earth, it will not be destroyed; | 土山不摧 |
| Formed by pure *qi*, | 一炁所成 |
| So its sound is difficult to trace. | 其音難追 |
| Divine animals of primordial origin, | 元始神獸 |
| The eight mighty dragons born from stones, | 石生八威 |
| Always close their eyes [to Introspect themselves]. | 長在瞑目 |
| The master rides thunder, | 夫子乘雷 |
| The perfected *qi* forms a halo, | 真炁圓生 |
| Golden rays fly all over heaven | 金光天飛 |
| So that the essence and energy flow out, | 以致流精 |
| This can keep you from fatigue by swallowing essence, | 漱挹不頹 |
| Letting you ascend to heaven of the Most High. | 能登太上 |
| It is only when death comes | 惟有死歸皆至 |
| That ignorant masses compete to go there. | 兢往愚兆 |
| Covering the grass with flowery lushness, | 華蕤上下 |
| Its mysterious sap, once exhausted, will regenerate; | 反生玄液 |
| It never withers, in summer or winter. | 不衰冬夏 |
| Grass has no branches or leaves. | 草之無枝 |
| Where is it hidden? | 其限在乎 |
| Jade rooms and golden ponds, | 玉房金池 |
| Sweet dew falls abundantly, | 霏霏甘露 |
| Purple clouds are floating, | 飄飄紫雲 |
| Fair figures ascending the rising summit, | 窈窕巍巍 |
| The jade girl climbing the mountain, | 所登玉女 |
| Is the Most High [immortal] in the right flow of the mountain, | 右流山之太上 |
| The form of Man-Bird. | 人鳥之形 |

The "palindromes" above are not in the reverse form of single words but in the reverse arrangement of phrases (two words). The form of the same palindromes can be found in Taoist texts from the Southern Dynasty (the same publication period as the *xuanlan* map), specifically the *Dongzhen taishang shangqing neijing* (洞真太上上清內經). The end of the text states that Shangqing zisu Yuanjun Shangzhen (上清紫素元君上真) wrote the *zhen* essay (真文) in the form of a palindrome ([Dongzhen taishang shangqing nei jing 洞真太上上清內經 1988](), p. 634). *Zhengdaoge*(證道歌) also said: "The sage taught his disci-

ples to cultivate Great-Dao. However, Great-Dao is so mysterious that it can't be expressed directly. Therefore, sages often write in palindromes" (教後學同歸大道, 玄妙豈可直言? 所以 聖人留訣意旨, 多作回文; Zuo 1988, p. 224). The text about the inner circle of the *xuanlan* map takes the form of a palindrome. It aims to show the airflow's direction of movement on the mountain: *left rotation* (左旋; counterclockwise) refers to the rise of earthly *qi*. In contrast, the text about the outer circle indicates that the mountain's airflow (the *qi* of heaven) follows the *right rotation* (右旋, clockwise); it means that *qi* is falling (see Figure 3).

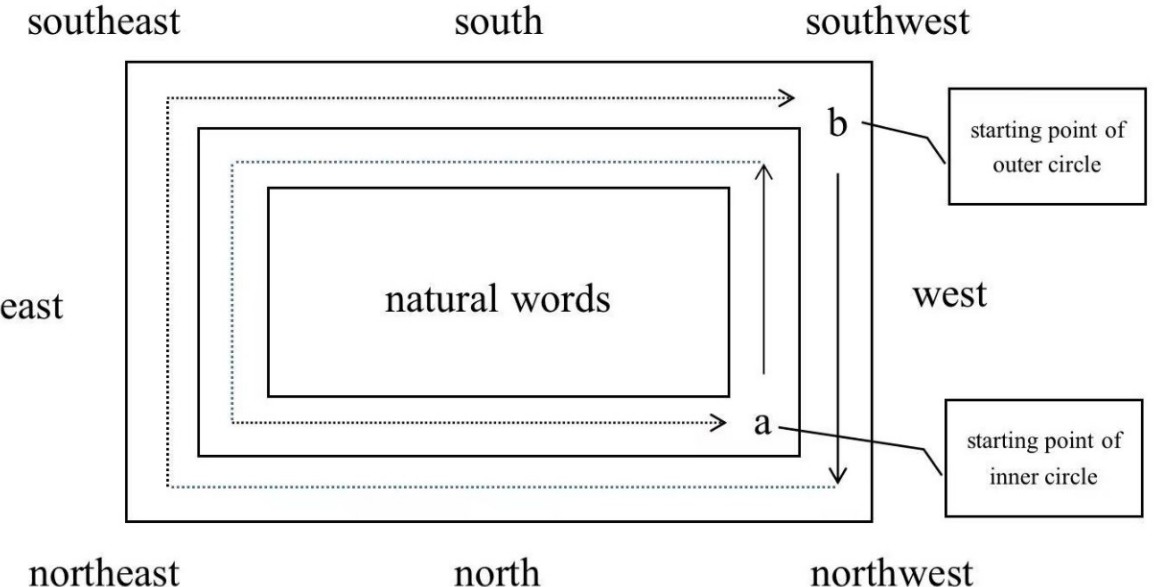

**Figure 3.** The text for following the direction of the inner and outer circles in Picture of the Mystic Vision (玄覽圖; author's drawing). Note: The text about the inner circle is read counterclockwise, starting from *Taishang* (太上) and followed by *Xianye* 仙也, so the number of words is 124. If read in a clockwise direction, the word *Yuanshi* (元始) must be removed; that is, 120 words. This is consistent with 124 + 120 = 244 words. [Picture was drawn by the author].

Hence, the characters in the void inscribed outside the mountain (山外空虛之字)—the inner and outer circles of the *Picture of the Mystic Vision* —mark the interactive movement of celestial and terrestrial *qi*, thus constituting a structural mechanism of combining yin and yang. This airflow movement mechanism explains the natural words inside the mountain— 山內自然之字—expressed by a string of proper nouns: *Arcuate Dragons moving* (弓龍行). What does *Arcuate Dragons* (弓龍) mean? This aspect will be explained later.

*3.2. Picture of the Primordial View (元覽圖)*

*Picture of the Primordial View* also includes text and images. The content of the scripture differs slightly from that of the *Picture of the Mystic Vision* but is generally the same. The images (TFCMB, 人鳥山形圖) are clearly different. There are no words around the circle of the *Picture of the Primordial View*, but there are horizontal "magic words" (符字) in the middle of the image (see Figure 4).

Huang Shishang and Luo Yiying both asserted that this line of "magic words" (符字) should be the *11 natural words inside the mountain* (山內自然之字) in the text of the *Picture of the Mystic Vision*: "[when] *Arcuate Dragons* move around [and] deities manifest themselves, dreadful calamities will be eliminated, and the ominous will be dispelled" (弓龙行, 神出, 除凶殃, 辟非祥) (Huang 2012, p. 152; Luo 2009, p. 115). In fact, the Taoist scripture of the Southern Dynasty, *Taishang qiuxian dingluchisu zhenjue yuwen* (太上求仙定錄尺素真訣玉文), has already revealed this "secret:"

> [when] *Arcuate Dragons* move around [and] deities manifest themselves, dreadful calamities will be eliminated, and the ominous will be dispelled. These 11

words are the secret spell of the Human-Bird Mountain. 弓龍行, 神出, 除凶殃, 辟非祥. 右十一字是人鳥山中符訣. ([Taishang qiuxian dinglu chisu zhenjue yuwen 太上求仙定錄尺素真訣玉文 1988](), p. 865)

It is very clear that the "magic words" (符字) in the middle of the *Picture of the Primordial View* are the *11 natural words inside the mountain*. The location is the junction between heaven and earth; that is, the intersection of *qi*, *yin*, and *yang* is between heaven and earth.

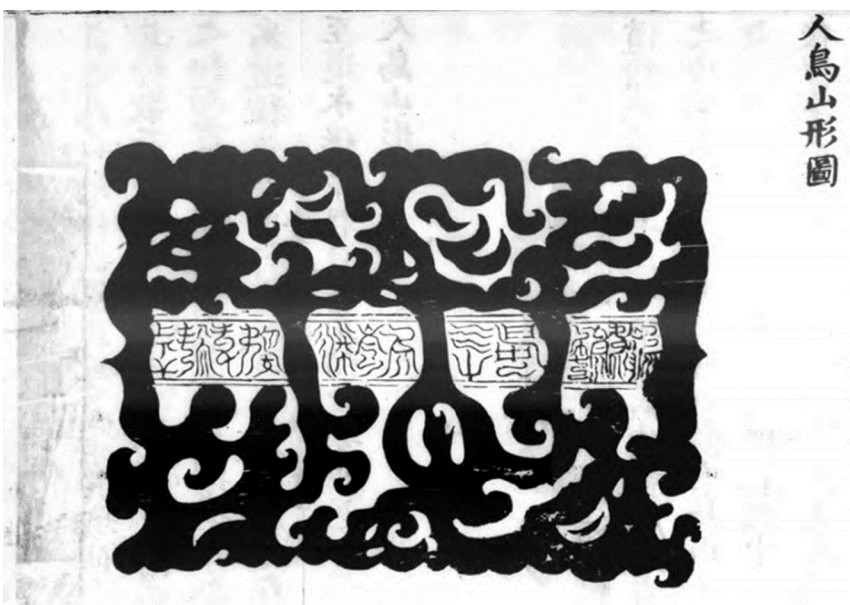

**Figure 4.** TFCMBM (Picture of the Primordial View, 元覽圖; [Primordial View of the Mountain Form of the Man-Bird Mountain 元覽人鳥山形圖 1445]()). [Photo reprinted with permission from *Zhengtong-daozang* 正統道藏, Official Publication of the Ming Dynasty, 10th year of the Zhengtong era. This picture has no copyright issues].

In order to understand the full significance of these 11 characters, the key is to grasp the meaning of *the Arcuate Dragons*. Confusingly, the *Arcuate Dragons* do not appear in other classical texts except in the two previously mentioned records. *Shiji fengshanshu* (史記·封禪書) records that a dragon drops its beard to welcome the ascension of Emperor Huang (黃帝). Some officials also wanted to follow the ascension; therefore, they grabbed the dragon's beard and Emperor Huang's bow, only to pull off the dragon's beard and drop Emperor Huang's bow to the ground. They did not go to heaven and were left holding the bow and the dragon's beard and crying (有龍垂胡髯下迎黃帝……餘小臣不得上, 乃悉持龍髯, 龍髯拔, 墮, 墮黃帝之弓. 百姓仰望黃帝既上天, 乃抱其弓與胡髯號; [Sima 1959](), p. 1394). This myth implies that the bow and dragon are symbols that lead people to heaven and make them immortal.

The *Image of the Han Dynasty* (漢畫) contains various *pictures of Arcuate Dragons* (弓龍圖). The so-called "*Arcuate Dragons*" (弓龍) are essentially graphic symbols expressing the mechanism of yin and yang. A typical example is displayed in Figure 5, which depicts the *Painting of a Bowed Dragon* from the Han Dynasty. The picture consists of two *arcuate dragons* forming a relatively closed space, inside which the tomb owner (soul) and driver ride a fish cart. Jiang Sheng (姜生) maintained that it is the Dragon Palace (龍宮), "implying the passage of the deceased from death (soul) through the interplay of yin and yang to the transformation process of regeneration into immortality (just like the womb of a pregnant woman)" ([Jiang 2016](), pp. 318–20). The two dragons, one yin and one yang form a merging of *qi* of the yin-yang mechanism to guide the tomb owner's soul toward the heavenly gate (the tomb door).

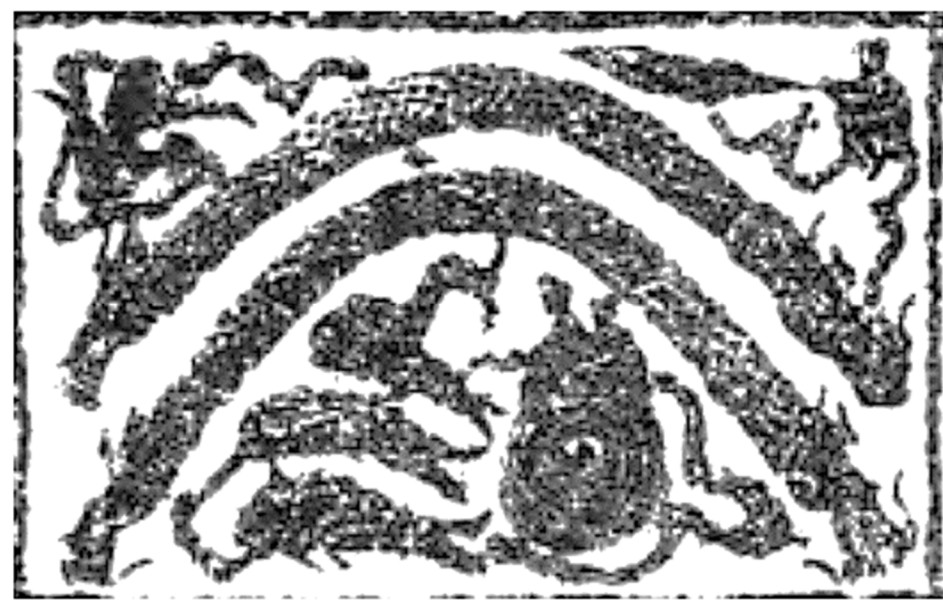

**Figure 5.** Arcuate Dragons pictured in a Han Dynasty painting (漢画弓龍圖, Jiang 2016, p. 320) [Photo reprinted with permission from Jiang sheng 姜生.].

　　Another representative image is the *Arcuate Dragons walking* (弓龍行) in the painting from the Han Dynasty. As seen in Figure 6, the tomb owner's soul rides on a fish cart, indicating that the soul is underground or in the local authority of the Northern Sea Water (北冥水府), with two fish mouths connected at the end of the cart and an arcuate dragon carrying it under the cart seat. This image suggests that the tomb owner's soul is in the process of "harmony between yin and yang" (*qi*; 陰陽合氣) so that the soul can be merged and resurrected from the dead. We can see that the theme of the *Arcuate Dragons walking* in the *Picture of the Primordial View* was produced according to the idea of "harmony between yin and yang" in the painting from the Han Dynasty.

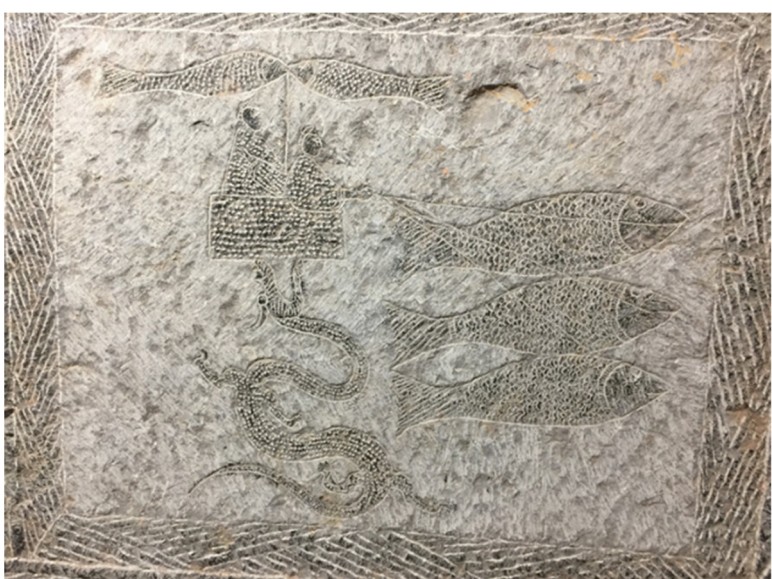

**Figure 6.** Arcuate Dragons are walking in a painting from the Han Dynasty (漢画弓龍行圖) (collected by the Xuzhou Han Dynasty Stone Portrait Museum, Jiangsu, China; photographed by the author on 15 July 2018). [This picture has no copyright issues].

　　However, the structural pattern of Han Dynasty paintings expressing the mechanism of "harmony between yin and yang" is more commonly presented as the *Dragon Cavern*

(龍穴) pattern formed by two dragons mating (which embodies "harmony between yin and yang", *qi*, 陰陽合氣). It is because the *Dragon Cavern* (龍穴) denotes the movement pattern of the two dragons' combination of *qi*: mating. The term *mating* originally referred to sexual intercourse among animals but was later extended to the universal combination of harmony between yin and yang (*qi*); mating represents yin and yang (交尾而表陰陽, Guo 1988, p. 473). As the image is "like a cavern" (see Figure 7), it is called the *Dragon Cavern* (龍穴).

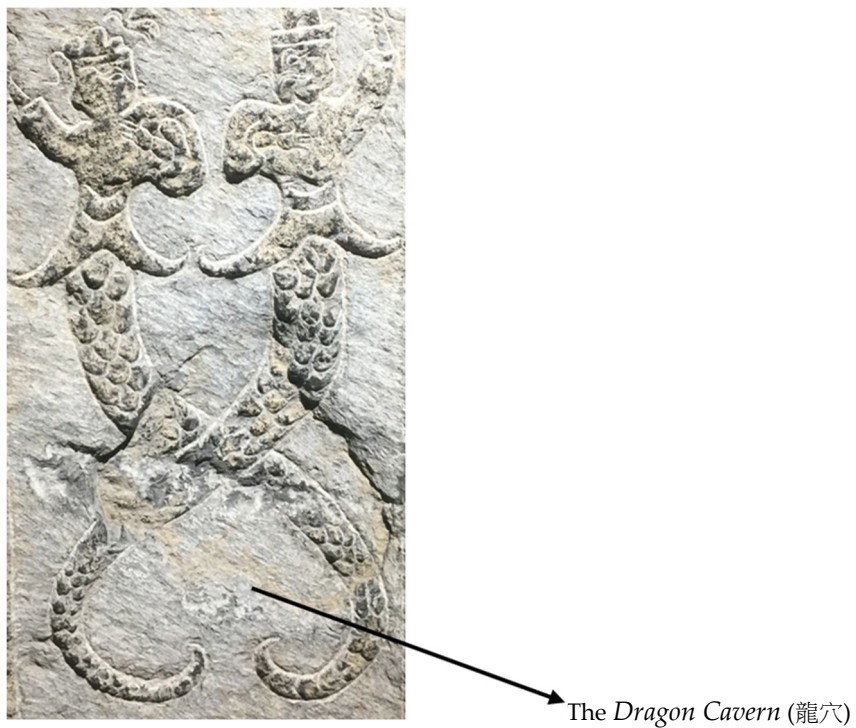

The *Dragon Cavern* (龍穴)

**Figure 7.** The mating of two dragons (雙龍交尾) and the *Dragon Cavern*, a Han Dynasty painting (collected by the Huaibei Museum, Anhui, China, photographed by the author on 14 July 2018). [This picture has no copyright issues].

Ancient Chinese geomancy regarded the "Dragon Cavern" as Fengshui Treasure Land (風水寶地). According to ancient feng shui masters, the "Dragon Cavern" is basically formed by the walking dragon (行龍結穴) or Arcuate Dragons (結弓龍). Hence, the shape of the Dragon Cavern is like a bow and arrow—" the hall is like a bow, the cave is like an arrow", "the hall is like a bow, the cavern is like an arrow that appears to be aimed straight towards the target" (堂似弓、穴似箭、朝似的, Guo 2017, pp. 171–72); it can store wind and accumulate *qi*. The "Dragon Cavern" is actually the cave shaped as a bow and arrow; that is, "Arcuate Dragons" in TFCMBM. Thus, according to the perceptions of ancient Taoists, TFCMBM was essentially a large "Dragon Palace" map filled with various "*Dragon Cavern*" symbols that symbolize the combination of yin and yang.

## 4. The Mechanism Inside the Body of the TFCMBM

As already mentioned, the TFCMBM implies a mechanism of "harmony between yin and yang", but where is it located? Is it an "immortal mountain" or "wonderland" that exists in a macrocosm (e.g., the Kunlun mountain chain)? Or does it refer to a metaphorical "immortal mountain" constituted by some part of the human body?

In this regard, we should pay attention to an important clue: the link between the texts of the inner and outer circles of the *Picture of the Mystic Vision* (玄覽圖) and the "two mountains" (Taiyuan and Changgu) described in the *Baopuzi neipian weizhi* (抱樸子內篇·微旨). Some scholars have found that the main content of the former was copied from the latter: "In the *Pao-P'u tzu* (抱樸子) these two passages are said to describe two different

mountains which 'he who seeks the Way must know' (求生之道，當知二山). The first is the Mountain of the Great Origin 太元之山，the second the Mountain of the Long Valley 長穀之山"(Lagerwey 1987, p. 166);" Both verses are based on passages in Chapter 6, 'Subtle meanings' [微旨], of the *Baopuzi*, in which Ge Hong discusses two mythical mountains: Grand Prime [太元] and the paradoxically named Changgu [長穀]" (Raz 2010, p. 1440). However, it is the breakthrough that we reveal regarding the TFCMBM.

The description of "two mountains" in *Baopuzineipian weizhi* (抱樸子內篇·微旨) is as follows (Ge 1988, p. 194):

| | |
|---|---|
| Mountain of Taiyuan, | 夫太元之山 |
| Difficult to know it, but easy to seek it, | 難知易求 |
| It is neither heaven nor earth, | 不天不地 |
| Neither sinking nor floating; | 不沉不浮 |
| Extremely dangerous and continuously following, | 絕險綿邈 |
| It is magnificent and ruggedly mountains. | 嶧嵬崎岖 |
| In the rising mist of harmonious *qi*, | 和氣絪縕 |
| The perfected is roaming. | 神意並遊 |
| The jade well is very quiet and deep, | 玉井泓邃 |
| It irrigates the body continuously. | 灌溉匪休 |
| One hundred and twenty celestial officers, | 百二十官 |
| Follow one another in the feudal official. | 曹府相由 |
| The *li* hexagram and the *kan* hexagram are distributed in order, | 離坎列位 |
| Ten thousand plants of mysterious ganoderma lucidum, | 玄芝萬株 |
| For scarlet trees to grow in particular, | 絳樹特生 |
| And their fruit are like pearls. | 其實皆珠 |
| The gold and jade are craggy, | 金玉嵯峨 |
| Sweet wine springs flow from crevices, | 醴泉出隅 |
| Gentlemen who have returned to youth, | 還年之士 |
| Fetch from its [the mountains] clean streams; | 挹其清流 |
| If you can practice, | 子能修之 |
| You become a companion of Fairy Song-Qiao. | 松喬可儔 |
| This is the first mountain. | 此一山也 |
| | |
| Mountain of Changgu, | 長穀之山 |
| Fair figures ascending the rising summit, | 杳杳巍巍 |
| Mysterious *qi* are floating, | 玄氣飄飄 |
| Liquor of immortality falls abundantly, | 玉液霏霏 |
| Golden ponds and purple rooms. | 金池紫房 |
| Are at the corners [acupoint] of the mountains. | 在乎其隈 |
| The ignorant people rush to go, | 愚人妄往 |
| But as soon as they are there, they will all die. | 至皆死歸 |
| Sensible people who know the Dao | 有道之士 |
| Can climb without crumbling. | 登之不衰 |
| Collect Polygonatum, and take it, | 采服黃精 |
| Hence he is able to fly in heaven. | 以致天飛 |
| This is the second mountain. | 此二山也 |

It is obvious that except for a few words, the description in the inner and outer circles of the text in the *Picture of the Mystic Vision* (玄覽圖) and the "two mountains" in *weizhi* (微旨) is the same: "the mountain of Taiyuan" (太元之山) corresponds to "the mountain of Taishang Man-Bird (太上人鳥之山; the text of the outer circle). Meanwhile, "mountain of Changgu" (長穀) corresponds to the *True Form of Taishang Man-Bird* (太上人鳥之形; the text about the inner circle). The objects shown in the *Picture of the Mystic Vision* (玄覽圖, including the *Picture of the Primordial View*, 元覽圖) are actually "two mountains" in *weizhi* (微旨). Thus, if we understand the meaning of the "two mountains" in *weizhi* (微旨), we will also be able to reveal the inner structure of the TFCMBM.

However, what is the meaning of the "two mountains" in *weizhi* (微旨)? Academics have been unable to solve this mystery. Li Fengmao (李豐楙) noted that "the two mountains are not to be sought from outside, but only to be found in the essence of thought"

([Li 2019](), pp. 317–18). However, he did not answer this question. Fu Daobin (傅道彬) suggested that Taiyuan mountain "symbolizes the male genital organs" and that Changgu mountain "symbolizes the female genital organs" ([Fu 1990](), p. 328). Yueli (朱越利) holds a similar view, arguing that the "two mountains" are a metaphor for men's and women's genital organs. The buttocks rise like a mountain peak, and the crotch can be regarded as the bottom of the mountain or the shadow of the mountain (山陰); therefore, they are both used as metaphors for the mountains, which are related to the Taoist view of intercourse (房中術; [Zhu 2007](), pp. 110–14).

Such perspectives, despite the lack of direct evidence and meticulous logical arguments, are not without enlightenment; it is certain that the "two mountains" really refers to the internal physiological mechanism of the human body. Hong (葛洪) has already hinted that the "two mountains" are not as conspicuously present on the earth as Mount Hua, Mount Huo, Mount Song, or Mount Tai. For people, the two mountains are "hard to know and easy to get" (難知易求). "Hard to know" (難知) means that the "two mountains" are not easy to see or discover because they are hidden inside the human body. This secret was kept by the ancient sages, who knew of it only by "precision thinking" (精思). "Easy to get" (易求) indicates that it is not necessary to travel long distances and take risks to reach the "two mountains", but rather to turn around and seek to reach them because the "two mountains" are within the human body ([Ge 1988](), p. 194). Su Dongpo (蘇東坡) once wrote a poem to remind the reader ([Su 1986](), p. 683):

The two mountains are very close to me, 二山在咫尺

The elixir is not the grass or trees. 靈藥非草木

Mysterious ganoderma lucidum is born in the mountains of Taiyuan, 玄芝生太元

Polygonatum flavum is produced in the mountains of Changgu. 黃精出長穀

Zhu (朱越利) claimed that "Su Dongpo [蘇東坡] understood the meaning of the two mountains" ([Zhu 2007](), pp. 110–14). However, what location do the "two mountains" indicate inside the human body? There is an important clue worthy of attention: there appears to be a subtle connection between *Baopuzineipian weizhi* (抱樸子內篇·微旨) and *suwen weizhidalun* (素問·微旨大論): (1) Both have the same title and are *weizhi* (微旨). (2) Both deal with the *qi*-flowing (氣流) mechanism inside the human body. (3) Both discuss the area of the *qi*-flowing (氣流) activity in the human body, which is related to the navel (shenque acupoint: 神闕穴) and its surrounding Changgu acupoint (長穀穴; *tianshu* acupoint: 天樞穴), the *taiyi* acupoint (太一穴, 太乙穴), the *guanyuan* acupoint (關元穴), and the large intestine meridian (大腸經脈). This correlation implies that the "two mountains" theory in *baopuzineipian weizhi* (抱樸子內篇·微旨) is derived from the "theory of the inter-induction of qi" (氣交論) in the *suwen weizhidalun* (素問·微旨大論). Thus, the "two mountains" theory is a theological description of the "theory of the inter-induction of qi" (氣交論). Although the present version of *suwen* (素問) was compiled by scholars in successive generations, the knowledge of meridian science originated from the pre-Qin dynasty, or even earlier ([Liu 1988](), pp. 50–59).

In this way, it is easy to see that the so-called "Changgu Mountain" (長穀之山) and "Taiyuan Mountain" (太元之山) in the *weizhi* (微旨) are related to the human body's internal meridian acupoints. In this system of meridian acupoints, Changgu (長穀) refers to the two acupoints of *changgu*, also known as *tianshu* (天樞), *changxi* (長溪), and *gumen* (穀門). They are located to the left and right of the navel (*shenque* acupoint: 神闕穴), 2 inches from the middle of the navel. *Weizhi* (微旨) refers to the Changgu Mountain (長穀之山) (the *True Form of the Taishang Man-Bird*) (太上人鳥之形) in *Picture of the Mystic Vision* (玄覽圖 cares about its *wei* [隈]; *wei* [隈] refers to the acupoints across the human body). The *changgu* acupoint (长谷穴) contains two paths of *qi* (氣) and blood movement: On one path, the *qi* and blood in the acupoint move along the stomach meridian. On the second path, *qi* and blood are collected and sent to the large intestine. The operating mechanism is such that the upper and lower parts of the stomach meridian converge at this acupoint. If the *qi* and blood are full, the essence, *qi*, can ascend to the large intestine meridian; that is, to

deliver *qi* and blood to the higher *tianbu* (天部; large intestine meridian, above the stomach meridian). Thus, the *changgu* acupoint (長穀穴) is also called the *tianshu* acupoint (天樞穴; Qu and Wang 2012, pp. 2–3), which is "the collection of the large intestine" and "the house of souls" (Xu 2014, p. 879).

Hence, this *qi*-and-blood operational mechanism has the function of promoting the internal clearing and turbidity of the human body and regulating *qi*: controlling the normal operations of internal digestion, absorption, and excretion. Ancient physicians linked the two acupoints of *changgu* into a line called the *tianshu* and stipulated that the lower part was the earth, and the upper part was heaven: "Heaven should be in the upper part, and the earth should be in the lower part. The acupoints are on both sides of the navel, and boundaries are between the upper and lower abdomen. They are connected to the *tri-jiao* [三焦] and can mediate the up and down…the duty is to function by ascending and descending" (上應之天, 下應之地, 穴當臍旁為上下腹之分界, 通于三焦, 有斡旋上下……職司升降之功) (Sun 2016, p. 519). As such, the *tianshu* marks the boundary line between heaven and earth in the human body, which is equivalent to the stomach meridian and is responsible for digesting food and coagulating *qi*. The ancient Taoists connected the two acupoints of the *changgu* and the large intestine meridian to form a triangle called the Changgu mountain (長穀之山), which means that the earthly *qi* (stomach *qi* and blood) rises to heaven (large intestine meridian; see Figure 8).

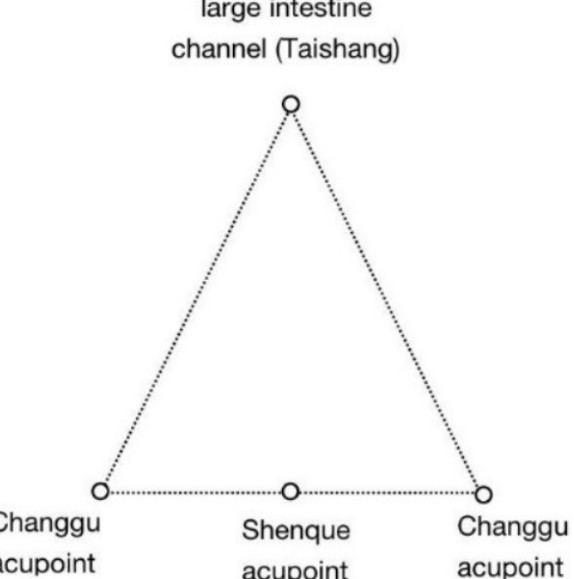

**Figure 8.** Structural diagram of the Changgu mountain (The reason why the large intestine meridian is called the Supreme Meridian is that it is located above the stomach meridian and *tianshu* 天枢). [Picture was drawn by the author].

The triangle in Figure 8 denotes the Changgu mountain (長穀之山); that is, the body's internal structural diagram of the *True Form of the Taishang Man-Bird* (太上人鳥之形) described in the inner circle of *Picture of the Mystic Vision*. This figure indicates the process mechanism of ascending *qi* and blood from the *tianshu* acupoints (stomach meridian) to the large intestine meridian. Therefore, the words "mysterious sap" 玄液 and "flowing essence" 流精 in the inner circle of the *Picture of the Mystic Vision* refers to the blood in the *tianshu* acupoint. A palindrome is used to express the upward path of *qi* and blood.

The word *taiyuan* (太元) should be a combination of the *taiyi* acupoint (太一穴; tai: 太) and the *guangyuan* acupoint (關元穴; yuan: 元). The *taiyi* acupoint (太一穴) is located in the upper 2–3 inches of the navel, 2 inches from the anterior midline (2 inches above the *tianshu* acupoint 天樞穴), one on each side. The operating mechanism of *qi* and blood in the *taiyi* acupoint (太一穴) is: "After the food is digested by the stomach, the essence generated is

gathered in the *taiyi* acupoint and thus delivered to all parts of the human body." Additionally, "the blood and *qi* of Large Intestinal Meridian of Foot Yangming 足阳明大肠经, which goes from the head to the foot, also runs to this acupoint and generates essential *qi*, which is the source of the acquired water cereal essence" (Zhu 2018, p. 119). In short, the *taiyi* acupoint (太一穴) is the place that transforms the blood of the large intestine meridian into an acquired essence, which is then transported downward to other parts of the body.

In the *Picture of the Mystic Vision*, the narration of "the essence of *qi* is like smoke (元炁煙熅; the rising mist of harmonious *qi* [和氣煙熅] in *weizhi* [微旨]), immortal is roaming (神真是遊; spiritual awareness is roaming [神意並遊] in *weizhi*) and so on, are in fact pictorial descriptions of the essence (*qi*) of movement in the *taiyi* acupoint. The *guangyuan* acupoint (關元穴), located 3 inches below the navel, belongs to the *ren* channel (任脈) meridian; therefore, there is only one acupoint called the *guangyuan* acupoint (關元穴) (Chen 1988, p. 493). According to ancient physicians, the *guangyuan* acupoint is the place where the body's vital energy is collected and developed, and "this acupoint is also called the Great Central Pole [大中極] because it is among the four sides of the human body; it is the place where men collect sperm and women store blood. The small intestine also collects. It is the place of the intersection of the foot three yin-yang and ren meridians. [此穴當人身上下四旁之中, 故又名大中極, 乃男子藏精, 女子蓄血之處. 小腸募也. 足三陰陽明任脈之會]" (Zhang 1965, p. 290). It means that the small intestine blood and *qi* gather at the *guangyuan* acupoint (關元穴), where they condense into the primordial *qi* (元氣) and are then transported to the skin. Moreover, "Jade liquid is abundant and pure" (玉液泓澄) and "the sweet spring comes out of the corner" (醴泉出隅) in the text from the *Picture of the Mystic Vision*, which portrays a vivid depiction of the state of essence (*qi*) running in the *guangyuan* acupoint.

Ancient alchemists placed great importance on the *guangyuan* acupoint (關元穴) as an important key point for "securing the essence and protecting the *qi*, not releasing the essence and *qi* randomly" (固精護氣, 不妄施泄) (Bai 1988, p. 853). "If your essence is kept in the *guanyuan* [關元], your *qi* will flow around; if the *qi* flows around, then your spirit will be full of vitality [神全]." (精既守于關元, 則氣周流; 氣周流, 則神全耳.) In ancient times, Shangqing Taoists connected the *taiyi* acupoint (太一穴) and the *guangyuan* acupoint (關元穴) to form another triangle: the Mountain of Taiyuan (太元之山; an inverted mountain), indicating that the *qi* of heaven (*taiyi* collects *qi* and blood from the small intestine) descends to the ground (the *guangyuan* acupoint, 關元穴; see Figure 9).

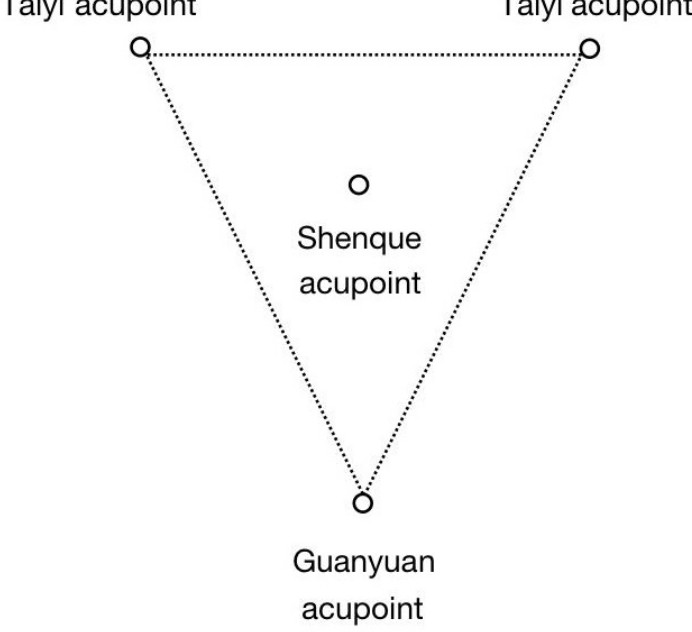

**Figure 9.** Structural diagram of the Mountain of Taiyuan. [Picture was drawn by the author].

In *Suwen·weizhi dalun* (素問·微旨大論), the integration of the two triangles in Figures 8 and 9 is the central area of the "convergence of celestial and terrestrial *qi*" (天地氣交) in the human body:

> At a place where heaven and earth meet, and the inter-induction of *qi* is live people. Hence, above the line of Tianshu, celestial qi is in charge, and under the line of Tianshu is terrestrial qi. The area of the inter-induction of qi is where the qi of human beings is generated, as well as all of creation. 上下之位, 氣交之中, 人之居也. 故曰: 天樞之上, 天氣主之; 天樞之下, 地氣主之; 氣交之分, 人氣從之, 萬物由之. (Zhang 2016, pp. 753–54)

The inner and outer circles of the text about *Picture of the Mystic Vision* and the "two mountains" in *Baopuzi neipian·weizhi* (抱樸子內篇·微旨) actually depict the area of the "convergence of celestial and terrestrial *qi*" (天地氣交) and its operating mechanism. It is the central mechanism of the human body to control the flow of water and food, raise clarity, and lower turbidity, and it is the place to condense and accumulate the essence. Thus, ancient Taoists regarded this area as a holy place for cultivation: "Man-Bird Mountain is the root of the universe and the cause of primordial pneuma." 是天地人之生根, 元氣之所因, 妙化之所用 (Xuanlan renniaoshan jing tu 玄覽人鳥山經圖 1988, p. 696). The TFCMBM, including the *Picture of the Mystic Vision* and the *Picture of the Primordial View*, form a sacred portrayal of the "convergence of celestial and terrestrial *qi*."

The structure of the meridians and acupoints shown in Figure 10 are the "two mountains" in the *Weizhi* (微旨), as well as the internal structure and operational mechanism of the human body, implied in the *Picture of the Mystic Vision* and *Picture of the Primordial View*. The 11 "words of nature in the mountain" in the *Picture of the Primordial View* are located on the *tianshu* between heaven and earth; it is the place where the soul of man and the spirit of the yin-yang combine: "*Arcuate Dragons*" (弓龍) (the dragon palace or dragon cavern). In the exquisite portrayal of ancient Taoists, the "*Arcuate Dragons*" are essentially the "true form of Man-Bird".

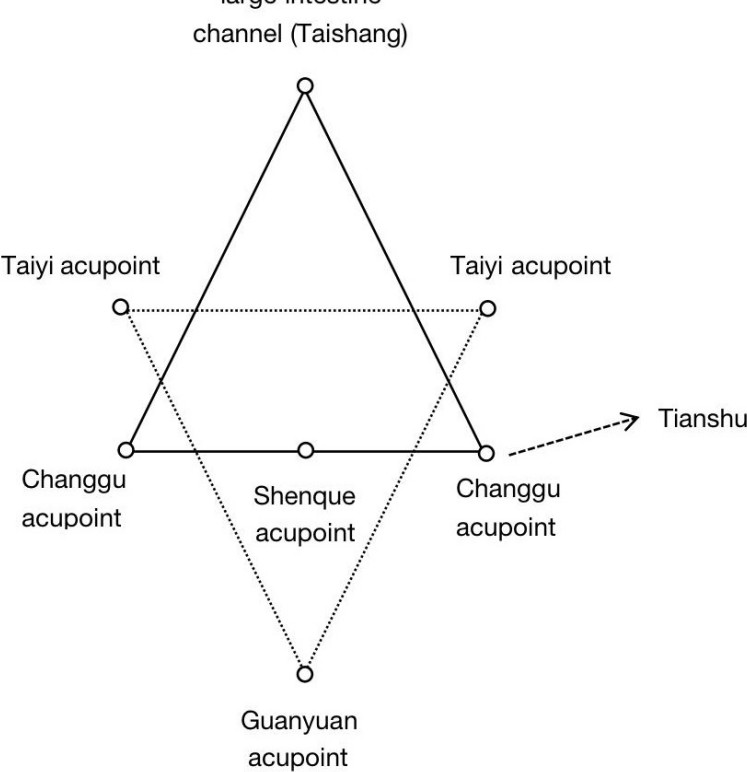

**Figure 10.** Structural diagram of the TFCMBM [Picture was drawn by the author].

Figure 11, located in the middle of *Picture of the Mystic Vision*, is a background opposing figure. Black is used as the background color, and the white part is a curved den, which indicates the "*Arcuate Dragons*" or "*Dragon Cavern*;" it also denotes the acupoints inside the human body. However, if white is used as the background color, then the black part of the white den is visible as an image of a bird's head and a human body, which is called "the true form of the Man-Bird" (人鳥真形). The implicit meaning of this chart is that the "*qi* of yin and yang" inside the human body, through the *Dragon Cavern* (龍穴; acupoints of the human body), are combined into the "the true form of the Man-Bird." It is also known as "essence-*qi*" (精氣) or "primordial *qi*" (元氣). Moreover, the "essence-*qi*" (精氣) is collected (i.e., Polygonatum flavum, flowing essence or mysterious light, 黃精、流精或玄輝) from the drug, which "leads to heavenly flight." Is this not the ultimate purpose of the ancient Taoists who drew the TFCMBM?

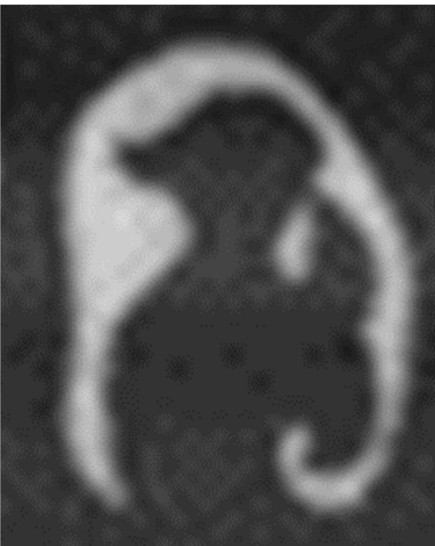

**Figure 11.** Form charts of the *Arcuate Dragons* (弓龍) and Man-Bird (*Picture of the Mystic Vision*,). [Xuanlan renniaoshan jing tu 玄覽人鳥山經圖 1445; Photo reprinted from *Zhengtongdaozang* 正統道藏, Official Publication of the Ming Dynasty, 10th year of the Zhengtong era. This picture has no copyright issues].

## 5. Ritual Functions of the TFCMBM

Data show that the TFCMBM, as ritual images, were mainly used for meditating spirit and activating *qi* (存神行氣) during rites. In this regard, the scripture of the *Picture of the Mystic Vision* makes it clear:

> Serving it respectfully, meditating and chanting it, accepting and sacrificing it, repairing and wearing it, practicing elaborately according to it, cultivating strictly on the basis of it. A long time without slacking, it can harmonize the three *qi*. 敬而事之, 存而念之, 受而醮之, 繕而帶之, 精而行之, 和而密之, 久無怠懈, 三氣調均. (Xuanlan renniaoshan jing tu 玄覽人鳥山經圖 1988, p. 696)

While sacrificing *jiao* (醮祭), the Taoists first put the TFCMBM on a "golden mirror" and then closed their eyes to keep thinking (存, to "see" an internal vision) about many immortals such as the Great Dao Lord in Most High (太上大道), Tianzun of the Ten Directions (十方天尊), the Primordial True King (元始真王), and the Great Emperor of Primordial *qi* on Man-Bird Mountain (人鳥山元氣生神太帝君; Xuanlan renniaoshan jing tu 玄覽人鳥山經圖 1988, p. 697). As Wang Haoyue (王皓月) discovered, the distinguishing feature of the "sacrifice of *jiao* of the Man-Bird Mountain" (人鳥山醮祭) is different from the traditional Taoist sacrifice of *jiao* (醮祭), which emphasizes the importance of continuing to meditate (存思; Wang 2014, p. 35). However, the root cause is that the TFCMBM forms a symbolic diagram reflecting the mechanism of the "inter-induction of *qi*" (氣交) within the human

body, which must be used to guide Taoists to practice the "meditation" (存思) outlook. It is because, in essence, "meditation" is the "inter-induction of *qi*" (氣交). Therefore, the TFCMBM essentially forms a picture of the "meditation" mode.

Historically, the period when ancient Taoists first used the TFCMBM in Taoist Praying Ceremonies (齋醮儀式) was probably the end of the Eastern Jin Dynasty or the beginning of the Southern Dynasty. Many Taoist classics of the Southern dynasties recorded the secret method of the "sacrifice of *jiao* of the Man-Bird Mountain" (人鳥山醮祭), including Dongxuan lingbao danshui feixing yundu xiaojie miaojing 洞玄靈寶丹水飛術運度小劫妙經 (1988, p. 859), *Dongzhen taishang cangyuan shanglu* (洞真太上倉元上錄; it is also known as the *Classic of the Man-Bird* [人鳥山经] written by Lu Xiujing [陸修靜]) (Zhang 2010, p. 71), and *Dongzhen taishang badao mingji jing* (洞真太上八道命籍經). The last one contains a paragraph of text on *The Mantra of the Man-Bird Mountain*" (人鳥山經咒). The mantra says: "To close your eyes and meditate, clicking your teeth five times, swallowing saliva five times, keeping five gods in your body…after a long pause, then, swallowing your saliva three times, clicking your teeth three times, reciting *the mantra of the Man-Bird Mountain*: Regulating *qi* in Central Yellow [黃中] and killing demons, and searching for and destroying the devil…"(思神閉目, 叩齒五, 咽液五, 存五神備在身中…良久, 咽液三, 叩齒三, 誦《人鳥山經咒》曰: 黃中策炁斬妖宗, 察制強精斷邪翁…, Dongzhen taishang badao mingji jing 洞真太上八道命籍經 1988, p. 512).

The meditating spirit and activating *qi* (存神行氣) form the core of the "sacrifice of *jiao* of Man-Bird Mountain" (人鳥山醮祭). All these activities, such as imagining deities (思神), closing the eyes (閉目), clicking the teeth (叩齒), swallowing liquid (咽液) and so on, are meant to help Taoists to conduct the "inter-induction of *qi*" (氣交) within the human body. According to *the mantra of the Man-Bird Mountain*, the internal movement mechanism of the "inter-induction of *qi*" (氣交) regulates *qi* in Central Yellow (黃中策炁).

Then, what is regulating *qi* in Central Yellow (黃中策炁)? Examining the literature，the term "Huang Zhong" first appeared in the *Wenyan Commentary to the Yijing*, in reference to Kun Hexagram (Wang and Kong 1999, p. 32):

A gentleman is regulating qi in Central Yellow, 君子黃中通理

To keep all parts of his body in the right position 正位居體

Harmony and beauty inside the body, 美在其中

And the essence-qi (精氣) flows freely through the limbs, 而暢於四支[肢]

Making career development. 發於事業

It is beautiful to the extreme. 美之至也

Although this passage expresses aConfucian moral, it clearly contains traces of early ancient Chinese knowledge of meridians. According to the *Su Wen* 素問, "The qi of the body's extremities all comes from the stomach …Therefore, the mechanism that delivers the stomach's bodily fluids controls the healthy balance of the limbs (四支(肢) 者, 皆稟氣於胃……故為胃行其津液者, 調衡四肢也; Xu 2019, p. 140)". In other words, the mechanism of generation and transportation of water and grain essence in the stomach meridian point (Central Yellow 黃中) is what makes the body's limbs nourished and energized. It is worth noting that the term "Central Yellow" (黃中), which was rarely used by ancient medical practitioners, was probably invented by Confucianists or Yin-Yang scholars during the Han Dynasty. They borrowed the term "Ecliptic" (黃道) from astronomy to name the meridian points of the human stomach.

A little later, some alchemical Taoists introduced the term "Central Yellow" (黃中) into the alchemical discourse. For example, the famous Han Dynasty alchemy classic *Zhou Yi Sen Tong Qi* (周易参同契) records that "Regulate qi in Central Yellow gradually, muscles and skin will moisturize"(黃中漸通理, 润泽达肌肤; Wei 1988a, p. 40). What's more, Peng Xia (彭晓), an alchemist in the Tang Dynasty, directly described the process of alchemy with a passage from the *Wenyan Commentar* (文言传) about "regulating qi in Central Yellow" (Peng 1988, p. 138). However, Taoist scholars of all dynasties did not publicly explain this concept to the outside world. In fact, they kept it a secret. Only Xue Yanggui (薛陽貴), a Taoist priest from the Qing Dynasty, delineated it in a little detail in the *Dialogue of Mei-*

*hua* (梅花問答編). He mentioned in his book that Central Yellow (黃中) is actually "ecliptic" (黃道), also known as the "Fairy Way" (仙道) in the *Classic of Ancient Alchemy* in Taoism. It is a passage inside the human body, where *qi* rises from the *guanyuan* acupoint to Kunlun (昆侖; the upper elixir field 上丹田), then successively drops to Jiangque (絳闕; the middle elixir field 中丹田), the elixir field (丹田; the lower elixir field 下丹田) and flows into *qi* acupoints (氣穴). There must be some relationship between Central Yellow (黃中) and the area of the "inter-induction of *qi*" (氣交). Some ancient Taoist classics already vaguely reveal the answer to this secret. For example, *Dongzhen taiwei huangshu jiutian balu zhenwen* (洞真太微黃書九天八錄真文,), a Taoist classic in the Eastern Jin Dynasty (東晉), uses a metaphor to describe the following:

> Regulating qi in Central Yellow by the Most High, which can purify the high
>
> qi of the Mysterious City of Nine Tiny, will make life never die. 太上黃中理氣,
>
> 九微玄城精既(溉) 太和, 主命永無逝. (Dongzhen taiwei huangshu jiutian balu zhenwen 洞真太微黃書九天八錄真文 1988, p. 563)

"Mysterious City of Nine Tiny" (九微玄城) points to *guanyuan* acupuncture, where the essence-*qi* is stored (關元之中, 男子藏精之所也, Zhang 1988, p. 76). According to the principle of meridian theory, the essence-*qi* in *guanyuan* acupuncture comes from the stomach meridian; that is, the line of *tianshu*, the area for the inter-induction of *qi*. Thus, Central Yellow refers to the line of *tianshu*. In line with it, "Regulating *qi* in Central Yellow" (黃中策炁) is just the "inter-induction of *qi*" (氣交).

*Supreme Master Seeks Immortals to Record ChiSu ZhenJue Jade Script* (太上求仙定錄尺素真訣玉文), a Taoist scripture in the Southern Dynasty, contains a poem about "Regulating-*qi* in Central Yellow" (黃中理氣, Lu 1988, p. 849):

> To regulate qi in Zhong Huang, 中黃理氣
>
> Primordial qi will be naturally generated, 元生自尊
>
> Make the stomach tube smooth, 胃管幽通
>
> And the six hollow [Fu] organs all are protected. 六腑俱存

Here, the "stomach tube" (胃管) refers to the stomach meridian (胃經); that is, the line of *tianshu* where the two acupoints of Changgu are located. The mechanism of "Regulating-*qi* in Central Yellow" aims to unblock the *qi* and blood of the stomach meridian and to protect the internal organs. Hence, the content of this poem is essentially a description of the mechanism of the "inter-induction of *qi*" (氣交). In this regard, *Yuqing wuji zongzhen wenchang dadong xianjing zhu* (玉清無極總真文昌大洞仙經注), an annotated version of Taoist scriptures in the Yuan Dynasty, describes the matter more clearly:

> It will protect your life for a long time if you can regulate qi in Zhong Huang and let the primordial Lord (元君, i.e., 元氣or the primordial spirit 元神) rule the universe of the body. The mechanism of human life is fundamentally in the digestion of grains [food in the body] by the spleen and stomach, which can nourish the Four Spirits [limbs, refers to the entire body]. 黃中治炁, 總統元君, 所以人之性命, 根本在乎脾胃克化穀府, 滋養四靈(四肢), 則性命可以長保. (Wei 1988b, p. 660)

This quote clearly indicates that the mechanism of "Regulating *qi* in Central Yellow" (黃中理氣) is to digest food, generate primordial *qi*, and nourish limbs (the entire body). In other words, "Regulating *qi* in Central Yellow" is essentially the "convergence of celestial and terrestrial *qi*" (天地氣交).

Interestingly, Li Xuanzhen (李玄真), a Taoist priest from the Tang Dynasty, wrote 32 magic words (see Figure 12) in *Shangqing jinmu qiuxian shangfa* (上清金母求仙上法):

> To regulate qi in Central Yellow, 黃中理氣
>
> Primordial qi will be naturally generated, 元生自尊
>
> Make the stomach tube smooth, 胃管幽通
>
> And the six hollow [Fu] organs are all protected. 六腑俱存

White Primordial [*qi*] Young Master 白元公子

And TaoKang Lord [*qi*] is nourished, 桃康得淳

Red elixir is brought up, also. 養育朱丹

Cave-Spirit [essence-*qi*] is moving back and forth in there. 洞神往返

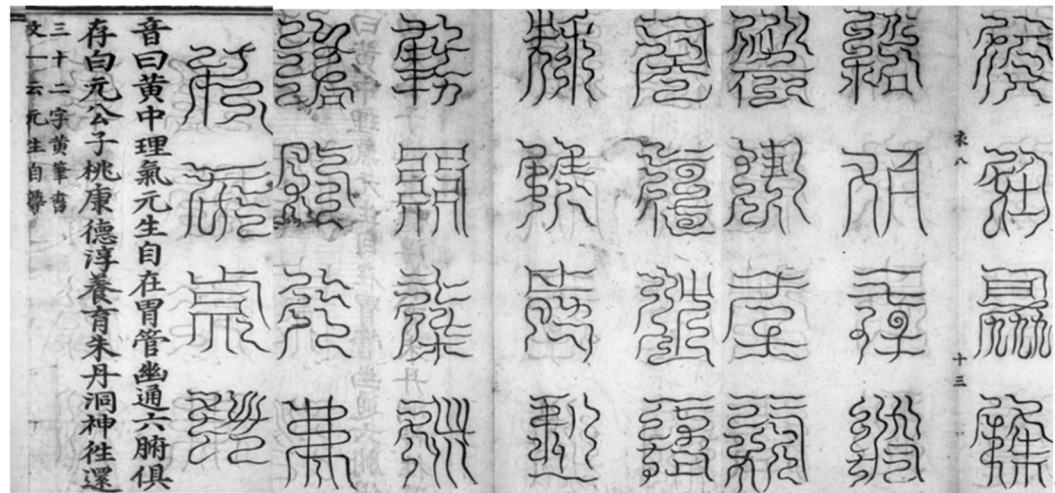

**Figure 12.** The 32 magic words (Li 1445). [Photo reprinted *Zhengtongdaozang* 正統道藏, Official Publication of the Ming Dynasty, 10th year of the Zhengtong era. This picture has no copyright issues].

The 32 magic words are a clearly condensed version of the *Picture of the Mystic Vision*, which succinctly depicts the scene of "Regulating *qi* in Central Yellow" in mythological vocabulary. In *Linbao lingjiao jidu jinshu* (靈寶領教濟度金書), the Song Dynasty Taoist Ning Quanzhen (宁全真) argues that the 32 magic words are a pithy formula for the *Picture of Eight Scenarios in the Lower Part of the [human body]*" (下部八景圖, Ning 1988, p. 771). Unfortunately, we did not find this picture in the book.

There are two pictures (see Figures 13 and 14) in which Central Yellow is drawn in *Xingming shuangxiu wanshen guizhi* (性命雙修萬神歸旨), a famous book about inner alchemy from the Ming Dynasty. They are titled *Puzhao tu* (普照圖) and *Zhongxin tu* (中心圖), respectively. Min Yide (閔一得), a famous Taoist priest from the Qing Dynasty, thought about the meaning of the two pictures (Min 1994, p. 686):

To regulate *qi* in Central Yellow, 黃中通理

Every part of the body is in the right state, 正位居體

Your *qi* links up between heaven and earth [in body]. 其氣即貫乎天地

The two pictures—which are the same as the *Picture of the Mystic Vision* (玄覽圖) and the *Picture of Eight Scenarios in the Lower Part of the [human body]* (下部八景圖)—also highlight the "inter-induction of *qi*" (氣交). Quite importantly, they clearly mark the position of Central Yellow in the human body.

As shown in Figure 13, Central Yellow refers to an acupoint that is parallel to Guizhong. The square in the figure is *Gui* [規]; it represents the earth, the navel, or the *shenque* acupoint, and its position is in line with the *changgu* acupoint on the right. Central Yellow refers specifically to the *changgu* or *tianshu* acupoint. However, Central Yellow also refers to the entire "line of *tianshu*." An inscription on the upper left side of Figure 13 says:

To regulate *qi* in Central Yellow, 黃中通理

Your body will become the valley of nothingness, 虛無之穀

The spring [essence-*qi*] caves of the Creator, 造化泉窟

And the dominator of the universe. 宇宙主宰

In the inscription, Central Yellow (黃中) is an apparent reference to the "line of *tianshu*", where there are about 27 acupoints and orifices of the human body from right to left:

Southwest Tower 西南樓——Zuqi acupoint 祖氣穴——Sarira 舍利子——Wuji Gate 戊己門——City of Buddha 法王城——Xuan Guan 玄關——Aerial 空中——Right-Orientation 正位——Real Soil 真土——Central Yellow 黃中——Ba-Bing 欛柄——Huang Ting 黃庭——Gui Zhong 規中——West 西方——Zhe-Ge 這箇 (primary cavity 祖竅)——Huang Po 黃婆——Zhong-Huang 中黃——Pure Land 淨土——Hun Kang 混康——Dan Jiong 丹扃——Altar for Concentration 守一壇——Wish-Fulfilling Pearl 如意珠——Orifices with Returning Root 歸根竅——Gateway for Rebirth 復命關——Void Hoard 虛空藏——Silent Sea 寂滅海——Hidden Brilliance 華光藏.

Although the *zhongxin tu* (中心圖) is brief, it also marks the position of Central Yellow (黃中). Two special words are inscribed on the picture: "Space between heaven and earth" (天地之間, left) and "Heart of heaven and earth" (天地之心, right). It indicates that Central Yellow (黃中) is the area connecting heaven and earth; that is, a place for the "convergence of celestial and terrestrial *qi*" (天地氣交_. Therefore, there are inscriptions on the right side of the figure:

> It is 84,000 li from the highest place in the sky to the lowest place on the earth…The human body is the same as heaven and earth. Therefore, Central Yellow [黃中] refers to the space between heaven and earth as [天地之間]…"Most harmonious primordial qi" [太和元氣] and "Greatest qi" [Noble Spirit, 浩然之氣]…all are born in Central Yellow [天之極上處至地之下極處, 總八萬四千里……人身亦然. 故曰: 天地之間, 亦曰黃中……而太和元氣, 浩然之氣, 皆由此中出也].

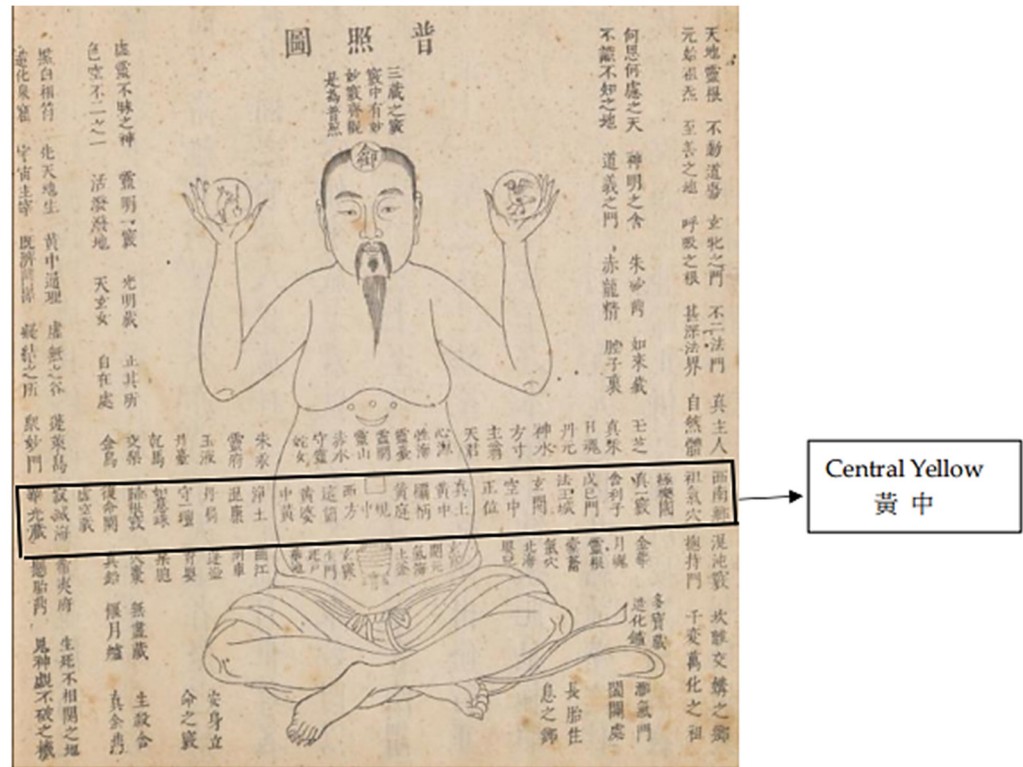

**Figure 13.** *Puzhao tu* (普照圖; Yin 1670, p. 21). [Photo reprinted from *Xingming shuangxiu wanshen guizhi* 性命雙修萬神歸旨, Di'etang 棣鄂堂, Publication in the 9th year of Kangxi 康熙 in the Qing Dynasty. This picture has no copyright issues].

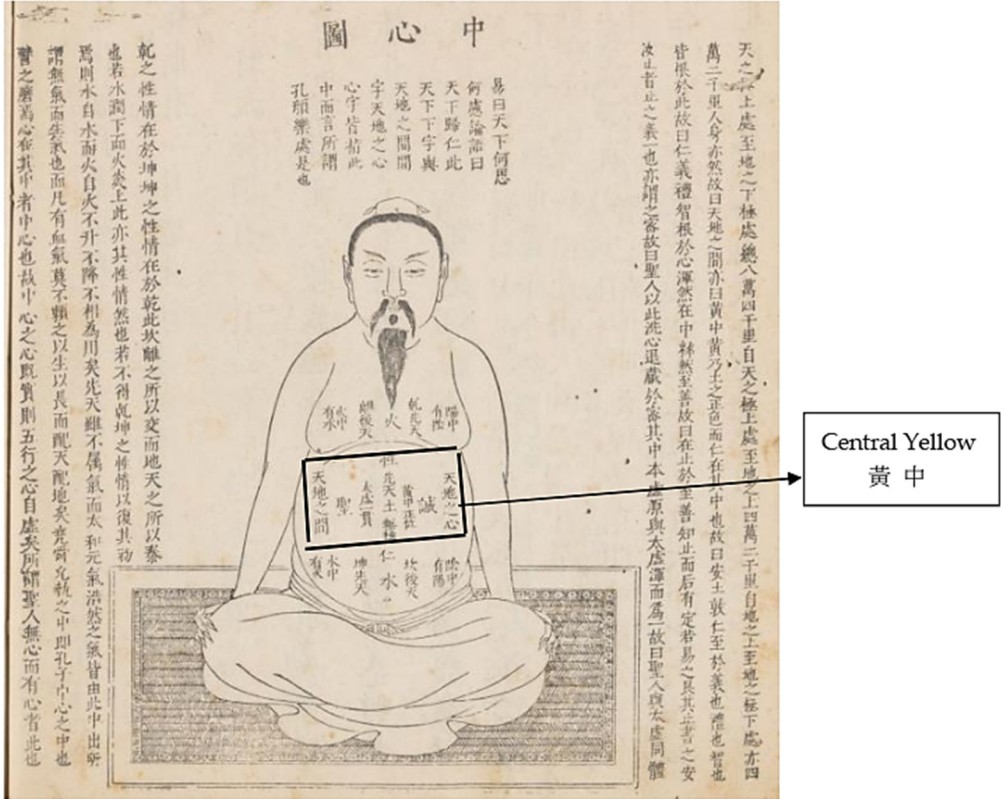

**Figure 14.** *Zhongxin tu* (中心圖; Yin 1670, p. 25). [Photo reprinted from *Xingming shuangxiu wanshen guizhi* 性命雙修萬神歸旨, Di'etang 棣鄂堂, Publication in the 9th year of Kangxi 康熙in the Qing Dynasty. This picture has no copyright issues].

This excerpt fully demonstrates that the ancient Taoists' so-called Central Yellow refers to the junction of heaven and earth inside the human body—the line of *tianshu*—which is the pivot of the convergence of celestial and terrestrial *qi* (天地氣交), mentioned in the *Su Wen* (素問). The core mechanism of the "inter-induction of *qi*" (氣交) is exactly regulating *qi* in Central Yellow (理氣). The main goal of the ancient Taoists in the process of practicing the "meditation" mode (存思) with the help of the TFCMBM was to obtain Polygonatum (黃精) or mysterious light (玄輝). Essentially, they are the same: the essence-*qi* or primordial *qi*. Polygonatum (Huang Jing [黃精]) refers to essence-*qi* that is generated and stored in Central Yellow (黃中). The metaphorical meaning of mysterious light (玄輝) is the brilliance of Huang Jing. Hence, ancient Taoists called it the "mysterious light of Huang Jing" (黃精玄暉), which is an upward (heavenly) flow of primordial *qi* and has the function of dispelling pent-up stress and cold, nourishing the stomach meridian, and making it smooth (黃精玄暉, 元陰上氣, 散鬱寒飆, 條靈斂胃, Linbao wuliang duren shangjing dafa 靈寶無量度人上經大法 1988, p. 745). Interestingly, according to the description of the ancient Taoists, Huang Jing (黃精) is not yellow but a red *qi* (赤氣). Red symbolizes yang (陽) and denotes positive life energy. When the human body is full of red *qi* (赤氣), Central Yellow (黃中), it is like a large warehouse full of grain (太倉) in which a newborn baby (赤子; natural self: 吾之自然) sits facing south (positive 陽direction) and eats the "grains", the red *qi* (赤氣, i.e., "mysterious light of Huang Jing" 黃精玄暉), and eventually becomes a real person, immortal (黃精赤氣填滿太倉中, 赤子當胃管中, 正南面坐, 飲食黃精赤氣…上為真人; Zhang 1988, p. 134). Clearly, in essence, this is exactly the mysterious picture described in *the Picture of the Mystic Vision* (玄覽圖) and *WeiZhi* (微旨). In short, the TFCMBM is essentially a theological description of the mechanism of the "inter-induction of *qi*" (氣交), and more specifically, regulating *qi* in Central Yellow within the human body.

In summary, the TFCMBM is not a theological description of an "immortal mountain" or "fairyland" far away from the world. Rather, it is a description of the real-life activities inside the human body. In fact, the ancient Taoists were able to draw a magical image because they had profound knowledge and technology of ancient Chinese meridians. It indicates that they had a special cultural characteristic: advocating for knowledge and technology and endowing them with sacredness. Hence, in ancient Taoism, the TFCMBM is ultimately regarded as the most sacred symbol to guide the Taoists to ascend to heaven and become immortal:

> If Taoists have the book Mountain Shape (the TFCMBM), they can become immortals, traveling and feasting on Kunlun Mountain…They do not need to take pills of immortality and practice the guiding technique. As long as the pure qi inside your body keeps generating, you will naturally ascend to heaven. 道士有此《山形》及書文備者, 便得仙度, 遊宴昆侖……不須服禦丹液, 無勞導引屈伸, 精之不休, 自獲升天矣. (Zhang 1988, p. 575)

## 6. Conclusions

We finally understand that the TFCMBM does not depict the "immortal mountain" far away from humankind itself but the landscape of life activity within their own bodies. From the perspective of epistemology, the particularity of the TFCMBM is that it depicts the internal life operation mechanism of the human body: the "inter-induction of *qi*" (氣交). The "knowledge" acquired by the ancient Taoists mainly came from their understanding of the "movement of *qi*" (氣運) in the human body. However, the nature of *qi* is that it is neither a concrete physical matter nor a metaphysical, spiritual entity, but a subtle life energy: "essence-*qi*" (精氣). Here, *jing* (精) refers to "subtle and minimum" (精微), which is almost equal to "nothing" and cannot be seen with the naked eye. However, it is also an "energy", a real force that constantly generates all things and drives them to move. In terms of its form of existence, *qi* is an ontological and intrinsic kinetic energy of life itself, which cannot appear in a realistic or concrete form. It means that *qi* and its movement cannot be expressed through an axiomatic or logical language symbol. It is the fundamental reason that ancient Taoists used theological metaphorical symbols to describe *qi* and its "movement."

In fact, according to the view of ancient Taoists, it is only correct to draw *qi* with fuzzy or chaotic image symbols. These "chaotic" (渾沌的) images and symbols are real true forms (真形). That is because the Tao (道 [trajectory]) of *qi* is essentially a "chaotic image" (渾沌之象), which symbolizes that life is in the process of eternal movement and change. Instead, it should be a deviation from the Tao if *qi* shows a clear and specific "shape", which means that *qi* stops its "natural movement;" that is, life is dead. As Zhuangzi (莊子) said:

> Supreme god riding on a light, making the shape of everything disappear without a trace, that is called "shining through the emptiness" (照曠). Everything is living according to its nature, heaven and earth are happy, and the shape of everything disappears, and everything returns to its emotional nature; that is called "chaos" (混冥). "上神乘光, 與形滅亡, 此謂照曠。致命盡情, 天地樂而萬物銷亡, 萬物 復情, 此之謂混冥". (Guo 1988, p. 438)

Thus, inevitably, the TFCMBM through which we "see" with the naked eye must be a "mass of gas" with a distorted shape and even fuzzy chaos rather than a geometric figure with a regular structure. In practice, the TFCMBM does not consist of images to "watch." In contrast, it is more like an "oracle" to guide you to give up "watching" to return to your "movement of *qi*" (氣運). Nevertheless, the "oracle" is not an admonition or revelation from someone else but the "natural spirit" (自然之神) inside your body, or more accurately, the essence-*qi* (精氣) naturally generated inside your body: the "information" and "knowledge" of your life conveyed to you. Interestingly, it is exactly the deep-rooted belief dogma of Taoists and even most Chinese people that the "movement of *qi*" (氣運) in your body is essentially your "destiny" (命運). In the Chinese context, the original meaning of "destiny" (命運) refers to life movement; that is, "movement of *qi*" (氣運). When you

are writing or reciting the TFCMBM, you are actually adjusting and improving your "destiny." Therefore, although it is difficult for us to describe the specific details of the ancient Taoists' writings of sacred knowledge, we already discern that their writing activities are, in essence, the practice of Tao (道), cultivating the Immortal True Form (真形).

**Funding:** This research was funded by the National Social Science Foundation of China: Research on the historical geography and cultural relics of the Twenty-four parishes in Taoism. Funder: National Office for Philosophy and Social Sciences, China. Fund Project Number: 22BZJ041.

**Conflicts of Interest:** The author declares no conflict of interest.

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
