# Peer review of "The Sacred Writing of Knowledge: Interpreting the True Form Charts of the Man-Bird Mountain in Taoism"

_religions, doi:10.3390/rel13111128_

Round 1

Reviewer 1 Report

This is an excellent research article. I recommend it for publication with great enthusiasm. 

In the same time, I would like to draw the Author's attention to several issues.

1. In my view, the List of References lacks an article by Prof. Lee Fong-mao “Grotto Heavens and Inner Realms: The Inner Visualization Meditations in Jiangnan Daoism from Second to Fourth Century” (李豐楙〈洞天與內景西元二至四世紀江南道教的內向游觀〉《東華漢學》92009157-197).

2. Address: page 24, lines 839-840.

Author’s text: Gil, Raz. 2010. Early Chinese Religion: Part Two: The Period of Division (220–589 AD). Ho Ping-Yü, Lu Gwei-Djen (collaborators). Leiden and Boston: Brill.

Clarifications: It seems that bibliographic record is not accurate. These two volumes (Chinese Religion: Part Two: The Period of Division (220–589 AD). 2 vols.) are edited by Prof. John Lagerwey and Prof. Lü Pengzhi. I don’t understand why Ho Ping-Yü and Lu Gwei-Djen are mentioned.

Why is there no title of the section, which was written by prof. Raz (Daoist sacred geography)? Or is it due to the rules adopted in the journal?

3. Address: page 24, line 847

Author’s text: Jiang, Sheng 薑生. 2016. Han diguo de yichan:…

Clarifications: As far as I know, the name of Prof. Jiang Sheng in all bibliographic databases, including Taiwanese, is written as 姜生. I also like the full forms of characters, but the form is usually considered not the full form of , but as its variation (異體字). Anyway, I’m afraid that an interested reader who wants to find other research works by Prof. Jiang in electronic bibliographic databases will not find anything by the name 薑生.

4. Address: page 25, line 848

Author’s text: Jiang, Sheng, and Tang, Weixia. 2010. Zhongguo daojiao kexue jishushi 中國道教科學技術史. Beijing: Kexue chubanshe.

My clarifications: a) Chinese characters are missing from the names of the respected editors. b) The full title of the book: 中國道教科學技術史: 南北朝隋唐五代卷. It’s only one of the volumes of a multi-volume project called “The History of Science and Technology in Taoism” (中國道教科學技術史).

4. Address: page 25, lines 852-854

Author’s text: Li, Fengmao 李豐楙. 2002. Duomian Wangmu, Wanggong, Kunlun, Donghua shengjingLiuchao Shangqing jing pai de fangwei Shenhua kaocha 多面王母、王公與昆侖、東華聖境———以六朝上清經派為主的方位神話考察. Taipei: Taiwan Zhongyang Yanjiuyan Zhongguo Wenzhe Yanjiusuo.

Clarifications: This outstanding research by Prof. Lee Fong-mao was published in a two-volume book: 李豐楙、劉苑如主編《空間、地域與文化 : 中國文化空間的書寫與闡釋》,上冊,臺北市 : 中央研究院中國文哲研究所2002Why is the source of this article not specified? Or is it related to the rules of referencing in the journal?

Author Response

Dear Reviewer1,

Please find my responses in the attachment. For more revised details, Please see the resubmitted manuscript.
Thank you very much for your valuable comments and suggestions, which corrected the errors in our manuscript and improved the academic quality of this paper.

Reviewer 2 Report

1) In order to have a complete view of the reference works on the subject, I suggest the author to integrate his remarks (citing it in the final Bibliography) with the recent article by Fabrizio Pregadio, "The Man-Bird Mountain: Writing, Prophecy, and Revelation in Early China", International Journal of Divination and Prognostication 2/1 (2020), pp. 29-82.

2) p. 2, line 26: Gil (2010) is wrong: must be Raz (2010).

3) p. 3, line 123: you cite a reference by G. Vitiello, but you write (Schipper 2002). You should write: (Vitiello 2004). Vitiello's short article is part of the collective work edited by K. M. Schipper and F. Verellen in 2004, not 2002.

4) p. 20, line 659: I do not understand well why you choose do not translate, here and elsewhere, the term Huangzhong 黃中. The use of a rendering like "Central Yellow" should be fine. Huangzhong is present in the Wenyan Commentary to the Yijing, in reference to Kun Hexagram: 君子黃中通理,正位居體,美在其中而暢於四支,發於事業,美之至也。One could also make some remarks on the probable derivation of the term Huangzhong from Huangdao 黃道, which initially indicated the ecliptic, i.e. the path taken by the sun, and later became a technical term especially in alchemy.

Author Response

Dear Reviewer2,

Please find my responses in the attachment. For more revised details, Please see the resubmitted manuscript.
Thank you very much for your valuable comments and suggestions, which corrected the errors in our manuscript and improved the academic quality of this paper.
